# Monocarboxylate Transporters 1 and 4 and Prognosis in Small Bowel Neuroendocrine Tumors

**DOI:** 10.3390/cancers14102552

**Published:** 2022-05-22

**Authors:** Niko Hiltunen, Jukka Rintala, Juha P. Väyrynen, Jan Böhm, Tuomo J. Karttunen, Heikki Huhta, Olli Helminen

**Affiliations:** 1Cancer and Translational Medicine Research Unit, Medical Research Center, Oulu University Hospital and University of Oulu, 90220 Oulu, Finland; jukka.rintala@ppshp.fi (J.R.); juha.vayrynen@oulu.fi (J.P.V.); tuomo.karttunen@oulu.fi (T.J.K.); heikki.huhta@oulu.fi (H.H.); olli.helminen@oulu.fi (O.H.); 2Surgery Research Unit, Medical Research Center, Oulu University Hospital and University of Oulu, 90220 Oulu, Finland; 3Department of Pathology, Central Finland Central Hospital, 40620 Jyväskylä, Finland; jan.bohm@ksshp.fi

**Keywords:** MCT1, MCT4, monocarboxylate transporter, lactate, neuroendocrine tumor, small bowel neuroendocrine tumor, immunohistochemistry

## Abstract

**Simple Summary:**

Monocarboxylate transporters (MCT) are cell membrane proteins transporting lactate, pyruvate, and ketone bodies. For most non-neoplastic cells, the aforementioned energy metabolites are waste products and must be exported via MCTs. However, it seems tumor cells have developed means to shuttle these metabolites through MCTs and utilize them for energy production. MCTs contribute to the acidification of the tumor microenvironment and are associated with drug resistance making them potential therapeutic targets. MCT expression has been shown to correlate with prognosis in a range of human solid tumors. In small bowel neuroendocrine tumors, the expression of MCTs 1 and 4 is reported for the first time. High MCT4 expression is associated with improved prognosis.

**Abstract:**

Monocarboxylate transporters (MCTs) are cell membrane proteins transporting lactate, pyruvate, and ketone bodies across the plasma membrane. The prognostic role of MCTs in neuroendocrine tumors is unknown. We aimed to analyze MCT1 and MCT4 expression in small bowel neuroendocrine tumors (SB-NETs). The cohort included 109 SB-NETs and 61 SB-NET lymph node metastases from two Finnish hospitals. Tumor samples were immunohistochemically stained with MCT1 and MCT4 monoclonal antibodies. The staining intensity, percentage of positive cells, and stromal staining were assessed. MCT1 and MCT4 scores (0, 1 or 2) were composed based on the staining intensity and the percentage of positive cells. Survival analyses were performed with the Kaplan–Meier method and Cox regression, adjusted for confounders. The primary outcome was disease-specific survival (DSS). A high MCT4 intensity in SB-NETs was associated with better DSS when compared to low intensity (85.7 vs. 56.6%, *p* = 0.020). A high MCT4 percentage of positive cells resulted in better DSS when compared to a low percentage (77.4 vs. 49.1%, *p* = 0.059). MCT4 scores 0, 1, and 2 showed DSS of 52.8 vs. 58.8 vs. 100% (*p* = 0.025), respectively. After adjusting for confounders, the mortality hazard was lowest in the patients with a high MCT4 score. MCT1 showed no association with survival. According to our study, a high MCT4 expression is associated with an improved prognosis in SB-NETs.

## 1. Introduction

Neuroendocrine tumors (NET) are neoplasms that can develop basically in any organ system where endocrine cells are present [1]. Prognosis varies in relation to anatomic site and tumor grade [2]. Gastroenteropancreatic NETs (GEP-NET) are among the most common ones, developing from the enteroendocrine cells of the gastrointestinal tract and pancreas [3,4]. The incidence rate of all GEP-NETs is 3.56 per 100,000 persons per year as reported by the National Cancer Institute’s Surveillance, Epidemiology, and End Results (SEER) Program. The incidence of small bowel NETs (SB-NET) is approximately 1.05 per 100,000 person-years [3]. The Surveillance of Rare Cancers in Europe (RARECARE) project estimated the European incidence of well-differentiated non-functioning GEP-NETs to be 11 per 1,000,000 person-years and 0.23 per 1,000,000 person-years for well differentiated functioning GEP-NETs [4]. NET incidence has increased 6.4-fold over the past decades, although this increase is at least partly due to increased number of upper gastrointestinal tract endoscopies and radiological imaging studies [3]. SB-NETs are among the ones with worst survival, but still due to the natural history of this neoplasia, deaths during Grade I differentiation occur during a very long follow-up with a median survival around 14 years [3]. The requirement of a long follow-up causes difficulties in the clinicopathological research of SB-NETs.

In most non-neoplastic cells, energy is mainly produced through aerobic mitochondrial metabolism, whereas in rapidly growing cells, such as cancer cells, energy is often produced by anaerobic glycolysis [5]. This results in high lactate and carboxylic acid levels [5]. pH regulation is therefore needed, and this is controlled in part by monocarboxylate transporters (MCTs) [6]. MCTs are a cell membrane transporter family encoded by SLC16 genes. The family comprises 14 different proteins, MCTs 1–4 being the most investigated ones [7]. MCT1 (SLC16A1) transports lactate into the cell but in some physiologic conditions it can contribute to lactate efflux. MCT4 (SLC16A3) is a key mediator in lactate efflux [8]. These MCT receptors have been associated with drug resistance, and are among potential therapeutic targets for treating solid tumors [9,10]. MCT1 inhibitors are in the advanced development phase and MCT4 inhibitors still in the discovery phase [11]. The expression patterns of MCT1 and MCT4 have been previously studied in many cancers, including gastric, lung, colorectal, and esophageal cancers [6,11,12,13,14]. However, according to our knowledge, no studies assessing the role of MCTs in neuroendocrine tumors or SB-NETs exists in clinical cohorts. A single study of mouse pancreatic NETs exist, which suggests that MCTs are upregulated in hypoxic conditions and may have a role in NETs as well [15].

In this study, MCT1 and MCT4 were studied for the first time in relation to SB-NET prognosis. MCTs have a significant role in the adaptation to acidosis. Acidosis, in turn, is suggested to contribute to cancer progression and aggressiveness. This suggests that MCTs could potentially affect prognosis, making MCTs 1 and 4 interesting research targets. The aim was to test the prognostic significance of MCT1 and MCT4 in SB-NETs in a large retrospective consecutive series of SB-NETs from two institutions in Northern and Central Finland with very long follow-up and complete disease-specific survival data.

## 2. Materials and Methods

### 2.1. Patients and Data Collection

The initial cohort and data collection has been previously described [16]. This study included 125 patients treated for ileal and jejunal SB-NETs in Oulu University Hospital and Central Finland Central Hospital. The patients in Oulu were treated between 9 February 2000 and 7 February 2018 and in Jyväskylä between 24 February 2000 and 31 December 2017. SB-NET lymph node metastases were detected in 95 of these patients and lymph node samples were also included in these cases. The collection of clinical patient data was possible through electronically stored archives. The Cause of Death Registry maintained by Statistics Finland provided the survival data. The end of follow-up was 31 December 2019. The Oulu University Ethics Committee approved this study (EETTMK 81/2008). The National Authority for Medicolegal Affairs (VALVIRA) waived the need for informed consent and approved for the use of data and samples.

Experienced pathologists determined the histological diagnoses at the time of treatment. The 8th edition of the AJCC/UICC TNM categories was used to determine tumor stage and the WHO 2019 classification of tumors of the digestive system was used to determine the tumor grade [17,18].

### 2.2. Immunohistochemical Staining

Tissue samples from primary tumors and lymph node metastases were fixed in formalin and embedded in paraffin at the time of diagnosis. Representative samples including those of the deepest tumor invasion were identified based on diagnostic hematoxylin–eosin slides. For immunohistochemistry, the sample blocks were retrieved from archives and cut into tissue sections of 3.5 µm in thickness. The sections were deparaffinized in xylene and rehydrated through graded alcohols. Antigen retrieval was performed with citrate buffer at pH 6 in a microwave oven, first at 800 W for 2 min and then at 150 W for 10 min. Tissue sections were cooled at room temperature for 20 min and rinsed both in distilled water and in phosphate-buffered saline containing Tween (PBS-Tween). Endogenous peroxidase activity was neutralized in peroxidase-blocking solution (Dako, Glostrup, Denmark, S2023) for 5 min, followed by two 5-min wash cycles in PBS–Tween. After this, sections were incubated with mouse monoclonal antibodies in dilute solution (Dako S2022); MCT1 for 60 min (diluted 1:100, Santa Cruz Biotechnology (H-1): sc-365501), MCT4 for 60 min (diluted 1:500, Santa Cruz Biotechnology (H-90): sc-50329). After another two 5-min wash cycles in PBS–Tween, samples were incubated with En-Vision polymer (Dako K5007) for 30 min and again washed in PBS–Tween for two cycles of 5 min. After the final wash, diaminobenzidine working solution (Dako K5007) was used as a chromogen. Lastly, the samples were rinsed in distilled water and counterstained in hematoxylin for 1 min. All staining was done with the Dako Autostainer (Dako, Copenhagen, Denmark). Cancer tissues with a high expression of MCT1 and MCT4 were used as external positive controls. To confirm the antigen preservation in the old paraffin blocks, we compared the MCT1 and MCT4 staining intensities in SB-NETs between old and new blocks divided by the median age of the blocks. No significant differences were found.

### 2.3. Immunostaining Assessment

Microscopic slides were scanned at 20× magnification using an Aperio AT2 digital slide scanner.

The immunoreactivity of MCT1 and MCT4 was analyzed by two independent researchers (JR and HH) who were completely blinded to the clinical data and did not participate in the statistical analyses. We assessed the intensity of staining (0–3), the percentage of positive cells (0–100), and the percentage of membrane-positive cells (0–100). The tumor stromal staining pattern was assessed as 0, no detectable staining; 1, focal staining; 2, areas of diffuse staining present in less than half of stromal area; 3, expression of moderate density distributed in more than half but not in all parts of the tumor stroma; 4, dense expression extending throughout the stroma as previously described [19]. Mean values of two independent estimates were used if there was no difference over 1 in the intensity or over 30% in the percentage. If the difference was more extensive, consensus was reached after re-evaluation. Re-evaluation was needed only in 3 cases. To avoid results found by chance or multitesting, predefined cut-offs for intensity (0–1 defined as low, and 2–3 as high), percentage (0–99 as low, and 100 as high), and stromal staining (0 as negative and 1–3 as positive) were used. Furthermore, based on both the intensity and percentage of positive cells, MCT1 and MCT4 scores were calculated as follows. A low intensity (0–1) was awarded zero points, and a high intensity (1–2) one point. Similarly, low percentages (0–99) were awarded zero points and high percentage (100) for one point. The intensity and percentage points were summed up, resulting in an MCT score of 0 (both low), 1 (low and high), or 2 (both high).

### 2.4. Statistical Analysis

For comparison of baseline values, The Chi-squared test for categorized variables and the Mann–Whitney U-test for continuous variables was used. Survival times were calculated from the date of operation until the time of death or the end of follow up. Disease-specific survival rates (DSS) were calculated using the Kaplan–Meier method with a log-rank test stratified by MCT intensity (low/high), percentage (low/high), stromal staining (low/high), and score (0, 1 and 2). Crude and adjusted hazard ratios (HR) for mortality were calculated using Cox regression models. Cox regression was adjusted for age, sex, stage (I-II, III, IV), the grade of differentiation (G1 or G2), adjuvant somatostatin therapy (no/yes) and sample type (surgical resection specimen or biopsy). A *p*-value of less than 0.05 was considered significant. The statistical analyses were performed with IBM SPSS statistics 27 for Windows (IBM Corporation, Armonk, NY, USA).

## 3. Results

### 3.1. Patients

Of the whole cohort, including 125 SB-NETs and 95 SB-NET lymph node metastases, adequate samples and representative immunostainings were available for 109 patients (MCT1) and 104 patients (MCT4). The final cohort included 61 lymph node metastases for both MCT1 and MCT4 evaluation.

Of the 109 SB-NET patients, total of 60 (55.0%) were men and the median age was 66 (IQR 56–72) years. Of tumors, 12 (11.0%) were stage I-II, 59 (54.1%) were stage III, and 38 (34.9%) were stage IV. Additionally, 85 (78.0%) were Grade I and 24 (22%) were Grade II whereas no Grade III tumors or neuroendocrine carcinomas (NEC) were operated for in either of the hospitals involved during the study period. Median follow-up time was 66 (IQR 40–119) months. Thirty-five patients died during follow-up, of which 20 deaths were considered to be disease-specific. Baseline characteristics are presented separately for patients with available MCT1 (Table 1) and MCT4 (Table 2).

### 3.2. MCT1 and MCT4 Staining

The MCT1 intensity median was 2.0 (IQR 1–2.75) and a total of 15 (13.8%) cases showed a staining intensity of 0. The MCT1 percentage median was 100 (IQR 100–100) with as much as 84 (77.1%) cases showing 100% staining. Examples of immunohistochemical staining are presented in Figure 1. MCT1 in tumor stroma was seen only in three cases and further analyses were not performed. All lymph node metastases showed negative MCT1 staining (even though external control was clearly positive).

The MCT4 intensity median was 1.0 (IQR 0–2.0) and 39 (37.5%) cases showed a staining intensity of 0. The MCT4 percentage median was 100 (IQR 0–100). A total of 74 out of 104 (71.2%) MCT4 cases were negative for stromal staining. Lymph node metastases showed MCT4 staining with MCT4 intensity median 2.0 (IRQ 1.0–3.0), percentage median 100 (IQR 100–100) and total of 41 out of 61 (67.2%) were negative for stromal staining. When comparing MCT4 expression between primary tumors and lymph node metastases, primary tumors showed both weaker staining intensity and percentage (both *p* < 0.001).

### 3.3. MCT1 and MCT4 Association with Clinicopathological Variables

Associations of the MCT1 score with clinicopathological variables are presented in Table 1. Only the tumor location in the jejunum showed a statistically significant association with a lower MCT1 score (*p* = 0.033). The MCT4 score showed no statistically significant associations with the studied variables (Table 2).

### 3.4. MCT1, MCT4, and Survival

Disease-specific survival in the whole cohort at 5 years was 91.2%, at 10 years it was 74.3%, and during the whole follow-up it was 64.8%. Overall survival at 5 years was 84.4%, at 10 years it was 61.3%, and during the whole follow-up it was 41.0%.

Neither the MCT1 intensity, percentage of positive cells, or the MCT1 score were associated with DSS (Table 3, Figure 2A).

A high MCT4 intensity was associated with better DSS when compared to low intensity (85.7 vs. 56.6%, *p* = 0.020) (Figure 2B). A high MCT4 percentage of positive cells resulted in a higher DSS when compared to a low percentage (77.4 vs. 49.1%, *p* = 0.059), although this was without statistical significance (Figure 2C). The MCT4 score of zero showed worse survival when compared with the MCT4 score of one and the MCT4 score of two (52.8 vs. 58.8 vs. 100%, *p* = 0.025) (Figure 2D). Survival percentages are presented in Table 3.

For future studies we aim to find the optimal cut-off values for MCT1 and MCT4 expression; we provided ROC curves of the MCT1 and MCT4 variables, as presented in Table 3. Optimal cut-offs based on the ROC curves showed only minor differences when compared to the currently used cut-off values (Appendix A).

Crude and adjusted HRs for disease-specific mortality are shown in Table 4. Adjusted HR in the high-MCT4-intensity group was 0.19 (95%CI 0.03–1.48) and in the high-MCT4-percentage group it was 0.23 (95%CI 0.09–0.61) when compared to low expression (Reference). MCT4 score showed a linearly lower adjusted mortality hazards with an increasing score (Score 1 HR 0.41; 95%CI 0.16–1.10 and Score 2 HR 0.00; 95%CI 0.00–0.00) when compared to Score 0 (Reference), although since no events occurred in the Score 2 group, HR could not be estimated.

## 4. Discussion

In this study, high MCT4 expression in SB-NETs was associated with an improved prognosis both in a univariate model and after adjusting for confounding factors. MCT1 expression was not associated with prognosis. In SB-NET lymph node metastases, MCT1 or MCT4 expression showed no prognostic value.

Our present study has several strengths. This study involved 109 patients, representing one of the largest published cohorts of SB-NETs with representative paraffin-embedded samples and complete clinical data. For the first time, MCT1 and MCT4 expression patterns were evaluated in SB-NETs, including 61 SB-NET lymph node metastases, and were examined in relation to clinicopathological variables and prognoses. We used whole section slides, including a selection of samples with the deepest invasion area. Our study included a long follow-up time, which is particularly important in relatively slowly progressing diseases like SB-NETs. Nearly 100% complete registry data with information on causes of death from Statistics Finland was used. This allowed for DSS analyses, which are of upmost importance in cancers with a good prognosis. Still, our study has some limitations. SB-NETs are rare neoplasms and despite including patients from a period of almost two decades, the sample size was relatively small, especially regarding the number of cancer deaths. Strong conclusions are possible only after replication studies. However, the association of high MCT4 expression and a good prognosis seems to be a real finding, with staining intensity, percentage, and score point estimates showing similar results both in uni- and multivariate models. According to the current analysis, mortality risk decreases with increasing MCT4 staining intensity and the percentage of positive cells, and future studies will need to set an optimal cut-off. Because of the long time period in the current study, the treatment and diagnostics of these tumors have possibly been improved, which can be a confounding factor. Although being non-selected for, the study population only consisted of G1 and G2 NETs, making data more homogenous and reliable. It should be noted that these results are not applicable for G3 NETs or NECs. A conservative approach has been favored in both study centers concerning G3 NETs and NECs as suggested in the ESMO guidelines [20].

SB-NET prognosis has mainly been determined by traditional factors such as Ki-67 proliferation index (Grades) and TNM stage. A nomogram including several different variables has also been developed by Modlin et al. in order to better predict the prognosis of individual patients [21]. At least three other studies have later tried to validate the nomogram for clinical use [22,23,24]. In addition to tumor-derived variables, continuous variables such as age, plasma CgA, and urinary 5-HIAA are used in the nomograms. Other clinical variables such as WHO performance status (PS) have been proved to be important in cancer surgery and PS probably affects SB-NET surgery outcomes similarly [25]. The validation study by Levy et al. found WHO performance status two to be an independent prognostic factor for a worse DSS in SB-NETs [24]. MCT4 has potential to improve the accuracy of survival prediction as it adds to the list of new emerging prognostic factors and biomarkers to predict disease progression and survival of SB-NET patients. MCT status could be included in these nomograms if the results of our current study are later replicated.

Acidosis, hypoxia, and glycolytic metabolism are associated with cancer cell viability and their ability to progress [26]. An intense proliferation of cancer cells results in hypoxia related to the growing distance to the microvasculature [27]. The resulting pH arrangement is characteristic for malignant tumors, resulting in extracellular acidification. The Warburg effect is a theory suggesting that cancer cells tend to switch to glycolytic metabolism even in normoxic conditions (aerobic glycolysis), producing large amounts of lactate [28]. To avoid intracellular acidification, lactate must be expelled to the extracellular space where lactate is not only a waste for tumors. It has been proposed that cancer cells become more aggressive and resistant to therapy in an acidifying microenvironment [29]. MCTs play a significant role in glycolysis and the adaptation to acidosis [5] and theoretically, according to the Warburg hypothesis, a high expression could be related to a poor prognosis, also making MCTs potential therapeutic targets [5].

MCT1 expression has been evaluated in several previous studies of certain human solid tumors. In a large cohort study of operated gastric cancer patients, MCT1 expression was associated with a higher T-class, but not with prognosis [12], and in esophageal adenocarcinoma, a high MCT1 expression was associated with an improved prognosis [6]. High MCT1 expression levels have been reported in, for example, breast and lung cancer [30,31]. In one gastric cancer study, an MCT1 inhibitor increased chemotherapy sensitivity, implying that targeting MCT1 may have therapeutic potential [9]. Still, no clear consensus in any solid tumor exists regarding MCT1 expression and prognosis. In SB-NETs, no previous studies exist. In the current study, the result seems to follow previous studies on several other solid tumors with no prognostic significance. It is noteworthy that despite adequate positive controls, lymph node metastases showed no MCT1 positivity, suggesting that only MCT1-negative clones were prone to metastatic behavior. Still, no survival difference in primary tumors was seen to be related to MCT1 expression.

In the aforementioned large gastric cancer cohort study, MCT4 was associated with positive lymph node status but again, not with prognosis [12]. In esophageal cancer, high MCT4 expression was associated with a poor prognosis [6], and similar results were found in previous studies on lung, colon, and prostate cancers [31,32,33]. Overall, in previous studies on solid tumors, MCT4 was commonly associated with a poor prognosis. Conversely to these findings, in the current study, which reports results for the first time in SB-NET, high MCT4 expression associated with improved survival.

The most studied function of MCT1 is in mediating lactate influx, however, in altered physiological conditions, MCT1 can also mediate lactate efflux [8]. MCT4 in turn is mostly expressed in highly glycolysis-dependent tissues where it mediates lactate efflux [8]. Lactate shuttling is an interesting phenomenon in cancers and various theories have been proposed to describe it [34,35,36,37]. This bi-directional shuttling of lactate seems to have an intricate role in the development, growth, and metastasis of cancer [38]. One of these theories, the reverse Warburg effect, theorizes that cancer cells can create a pseudo-hypoxic microenvironment in the stroma, inducing MCT4 upregulation and increased glycolysis in stromal fibroblasts through HIF-1α activation. The fibroblasts then efflux lactate through MCT4, after which lactate is imported by tumor cells through MCT1 and used as an oxidative metabolite [34]. Through MCT1 or MCT4 inhibition, this pathway could potentially be interrupted, leading to cancer cell starvation.

NETs are among the most extensively vascularized cancers, with an intratumoral vessel density approximately 10-fold higher when compared with carcinomas. Intratumoral microvascular density is the highest in low-grade tumors and it is associated with an improved prognosis and prolonged survival [39]. GEP-NETs express several proangiogenic factors including vascular endothelial growth factor (VEGF), which is the main driver of angiogenesis [40]. VEGF expression is strictly modulated by oxygen levels through HIF-1α [41]. Considering these GEP-NET-specific features, it seems plausible that increased hypoxia and acidotic tumor microenvironments could cause an increased activation of proangiogenic factors further increasing intratumoral microvascular density. Since the majority of our cases were G1, with some G2 NETs, this could cause favorable prognosis effects and could be one of the mediating factors explaining the high MCT4 activity and good prognosis.

The current study has some potential clinical implications. MCT proteins are potential targets for therapeutic purposes. Cancer cells are known to become more aggressive and treatment-resistant in an acidifying tumor microenvironment. Because of their role in controlling the acidity of the tumor microenvironment, MCTs could be important factors in regulating cancer progression. Understanding how MCT expression is tied to patient prognosis can help develop targeted therapies to slow down cancer progression or even therapies that lead directly to cancer cell starvation and cell death. Our study provides important information on the prognostic value of MCT1 and MCT4 in SB-NETs, showing the possibility of contrary results (regarding MCT4 expression) and heterogeneity between different cancers. Replication studies are, however, needed.

## 5. Conclusions

In our study, we reported the presence of MCT1 and MCT4 in SB-NETs. High MCT4 expression was associated with improved survival also after adjusting for confounders.

## Figures and Tables

**Figure 1 cancers-14-02552-f001:**
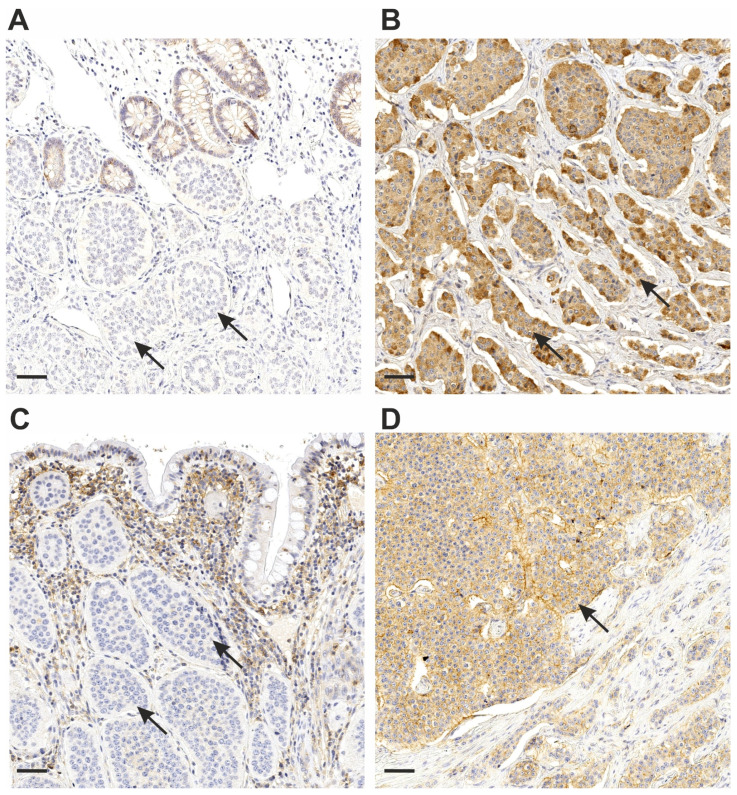
Immunohistochemical staining of MCT1 and MCT4 in representative small bowel neuroendocrine tumor samples showing (**A**) a low MCT1 score, (**B**) a high MCT1 score, (**C**) a low MCT4 score and (**D**) a high MCT4 score expression. The scale bar length is 50 μm (bottom left corner). Arrows indicate MCT-negative (**A**,**C**) and MCT-positive (**B**,**D**) tumor cells.

**Figure 2 cancers-14-02552-f002:**
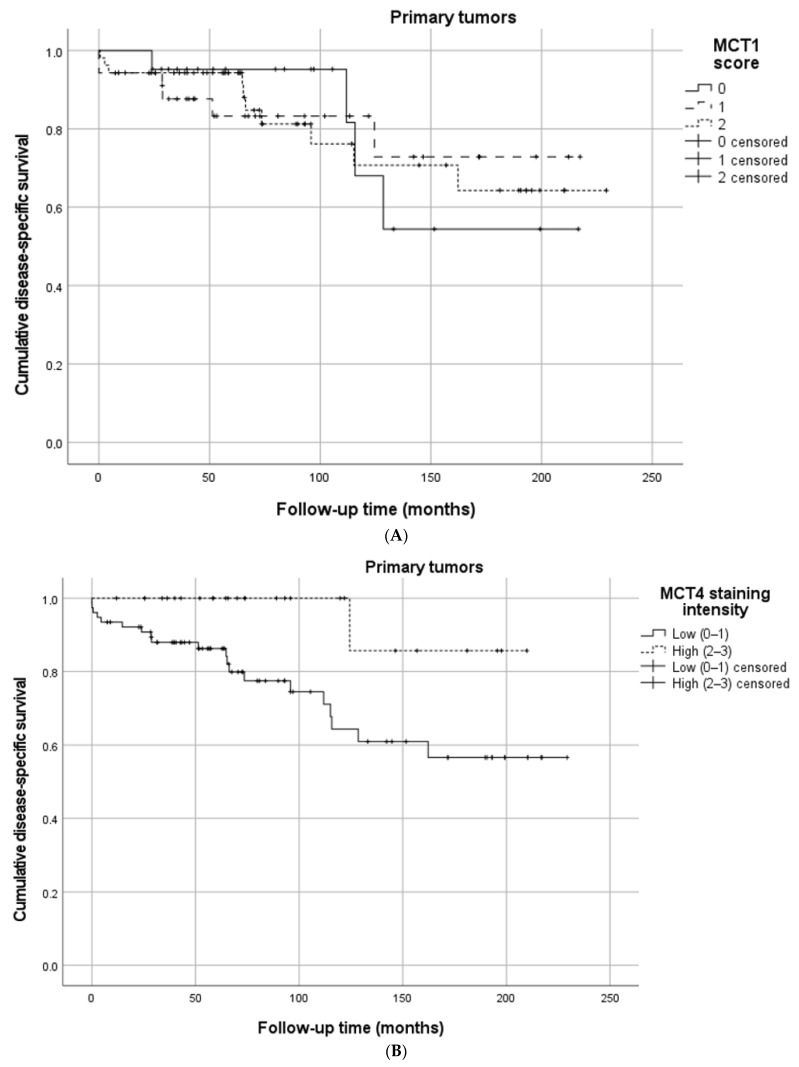
Cumulative disease-specific survival in primary small bowel neuroendocrine tumors stratified by (**A**) MCT1 score, (**B**) MCT4 staining intensity, (**C**) MCT4 percentage of positive cells, and (**D**) MCT4 score.

**Table 1 cancers-14-02552-t001:** Baseline characteristics and comparison stratified by monocarboxylate transporter (MCT) 1 score.

Variables	MCT1 Score 0, *n* = 21	MCT1 Score 1, *n* = 35	MCT1 Score 2, *n* = 53	P between Groups
**Sex**				
Male, *n* (%)	12 (57.1)	21 (60.0)	27 (50.9)	0.689
**Age, median (IQR)** years	61 (56–69)	66 (56–73)	66 (56–73)	0.866
**T-Class**				0.109
T1–2, *n* (%)	3 (14.3)	9 (25.7)	15 (28.3)	
T3, *n* (%)	13 (61.9)	17 (48.6)	16 (30.2)	
T4, *n* (%)	5 (23.8)	9 (25.7)	22 (41.5)	
**N-Class**				0.530
N0, *n* (%)	3 (14.3)	5 (14.3)	12 (22.6)	
N1–2, *n* (%)	18 (85.7)	30 (85.7)	41 (77.4)	
**M-Class**				0.860
M0, *n* (%)	13 (61.9)	24 (68.6)	34 (64.2)	
M1, *n* (%)	8 (38.1)	11 (31.4)	19 (35.8)	
**Stage**				0.950
I-II, *n* (%)	2 (9.5)	5 (14.3)	5 (9.4)	
III, *n* (%)	11 (52.4)	19 (54.3)	29 (54.7)	
IV, *n* (%)	8 (38.1)	11 (31.4)	19 (35.8)	
**Grade**				
1, *n* (%)	17 (81.0)	28 (80.0)	40 (75.5)	0.825
2, *n* (%)	4 (19.0)	7 (20.0)	13 (24.5)	
**Tumor location**				0.033
Ileum, *n* (%)	18 (85.7)	32 (91.4)	53 (100)	
Jejunum, *n* (%)	3 (14.3)	3 (8.6)	0	
**Somatostatin analogue treatment**				0.391
Yes, *n* (%)	12 (57.1)	16 (45.7)	32 (60.4)	
**Chemotherapy**				0.818
No, *n* (%)	16 (76.2)	28 (80.0)	45 (84.9)	
Preoperative, *n* (%)	1 (4.8)	2 (5.7)	2 (3.8)	
Postoperative, *n* (%)	4 (19.0)	4 (11.4)	4 (7.5)	
**Multiple primary tumors**				0.403
Yes, *n* (%)	8 (40.0)	8 (22.9)	15 (28.8)	
**P-CgA**				
≥3 nmol/L, *n* (%)	16 (88.9)	24 (85.7)	43 (87.8)	0.945
Median (IQR) nmol/L	8.5 (4.7–15.0)	6.4 (3.6–13.5)	4.8 (3.4–15.0)	0.587
**dU-5-HIAA**				
≥42 µmol/L, *n* (%)	10 (66.7)	18 (69.2)	21 (48.8)	0.193
Median (IQR) µmol/L	56 (37–199)	96 (27–149)	40 (25–169)	0.632

P-CgA, plasma chromogranin A; dU-5-HIAA, 24 h urine hydroxyindoleacetic acid.

**Table 2 cancers-14-02552-t002:** Baseline characteristics and comparison stratified by MCT4 score.

Variables	MCT4 Score 0, *n* = 46	MCT4 Score 1, *n* = 32	MCT4 Score 2, *n* = 26	P between Groups
**Sex**				
Male, *n* (%)	28 (60.9)	14 (43.8)	14 (53.8)	0.329
**Age, median (IQR)** years	62 (56–72)	69 (56–75)	65 (47–70)	0.258
**T-Class**				0.713
T1–2, *n* (%)	11 (23.9)	7 (21.9)	9 (34.6)	
T3, *n* (%)	19 (41.3)	16 (50.0)	9 (34.6)	
T4, *n* (%)	16 (34.8)	9 (28.1)	8 (30.8)	
**N-Class**				0.308
N0, *n* (%)	10 (21.7)	6 (18.8)	2 (7.7)	
N1–2, *n* (%)	36 (78.3)	26 (81.3)	24 (92.3)	
**M-Class**				0.948
M0, *n* (%)	29 (63.0)	19 (59.4)	16 (61.5)	
M1, *n* (%)	17 (37.0)	13 (40.6)	10 (38.5)	
**Stage**				0.959
I–II, *n* (%)	6 (13.0)	3 (9.4)	2 (7.7)	
III, *n* (%)	23 (50.0)	16 (50.0)	14 (53.8)	
IV, *n* (%)	17 (37.0)	13 (40.6)	10 (38.5)	
**Grade**				
1, *n* (%)	35 (76.1)	27 (84.4)	18 (69.2)	0.389
2, *n* (%)	11 (23.9)	5 (15.6)	8 (30.8)	
**Tumor location**				0.203
Ileum, *n* (%)	42 (91.3)	32 (100)	25 (96.2)	
Jejunum, *n* (%)	4 (8.7)	0	1 (3.8)	
**Somatostatin analogue treatment**				
Yes, *n* (%)	24 (52.2)	20 (62.5)	16 (61.5)	0.596
**Chemotherapy**				0.669
No, *n* (%)	35 (76.1)	27 (84.4)	24 (92.3)	
Preoperative, *n* (%)	3 (6.5)	1 (3.1)	0	
Postoperative, *n* (%)	6 (13.0)	3 (9.4)	2 (7.7)	
**Multiple primary tumors**				
Yes, *n* (%)	15 (33.3)	1 (35.5)	4 (15.4)	0.187
**P-CgA**				
≥3 nmol/L, *n* (%)	34 (91.9)	25 (86.2)	21 (84.0)	0.609
Median (IQR) nmol/L	7.6 (4.1–39.5)	5.0 (3.4–10.2)	5.5 (3.3–12.0)	0.209
**dU-5-HIAA**				
≥42 µmol/L, *n* (%)	19 (59.4)	15 (55.6)	15 (68.2)	0.658
Median (IQR) µmol/L	74 (32–266)	44 (24–104)	58 (27–222)	0.258

P-CgA, plasma chromogranin A; dU-5-HIAA, 24 h urine hydroxyindoleacetic acid.

**Table 3 cancers-14-02552-t003:** Disease-specific survival rates based on MCT1 and MCT4 stainings (intensity, percentage, MCT score, and stromal staining) in both primary tumors and lymph node metastases. Lymph node metastases showed negative MCT1 staining and therefore these are not presented.

Disease-Specific Survival	No. of Patients	MCT1 Intensity, Low	MCT1 Intensity, High		*p*
Primary tumors	109	66.7%	64.0%		0.825
	No. of patients	MCT1 percentage, low	MCT1 percentage, high		*p*
Primary tumors	109	51.8%	68.0%		0.841
	No. of patients	MCT1 score 0	MCT1 score 1	MCT1 score 2	*p*
Primary tumors	109	54.4%	72.9%	64.3%	0.999
	No. of patients	MCT4 intensity, low	MCT4 intensity, high		*p*
Primary tumors	104	56.6%	85.7%		0.020
Lymph node metastases	61	62.6%	73.6%		0.345
	No. of patients	MCT4 percentage, low	MCT4 percentage, high		*p*
Primary tumors	104	49.1%	77.4%		0.058
Lymph node metastases	61	71.1%	65.1%		0.856
	No. of patients	MCT4 stroma, negative	MCT4 stroma, positive		*p*
Primary tumors	104	65.7%	51.9%		0.469
Lymph node metastases	61	65.7%	75.6%		0.659
	No. of patients	MCT4 score 0	MCT4 score 1	MCT4 score 2	*p*
Primary tumors	104	52.8%	58.8%	100%	0.025
Lymph node metastases	61	70.0%	56.5%	73.4%	0.678

**Table 4 cancers-14-02552-t004:** Hazard ratios (HR) for disease-specific mortality with 95% confidence intervals (CI) in primary small bowel neuroendocrine tumors stratified by MCT4 intensity (low/high), MCT4 percentage (low/high), and MCT4 score.

**MCT4 Intensity**				
**Primary tumors**	**No. of patients**	**Low** **HR (95%CI)**	**High** **HR (95%CI)**	
Crude	104	1.00 (reference)	0.13 (0.02–0.99)	
Adjusted	104	1.00 (reference)	0.19 (0.03–1.48)	
**MCT4 percentage**				
**Primary tumors**	**No. of patients**	**Low, HR (95%CI)**	**High, HR (95%CI)**	
Crude	104	1.00 (reference)	0.44 (0.18–1.05)	
Adjusted	104	1.00 (reference)	0.23 (0.09–0.61)	
**MCT4 score**				
**Primary tumors**	**No. of patients**	**Score 0, HR (95%CI)**	**Score 1, HR (95%CI)**	**Score 2, HR (95%CI)**
Crude	104	1.00 (reference)	0.87 (0.36–2.06)	0.00 (0.00–0.00)
Adjusted	104	1.00 (reference)	0.41 (0.16–1.10)	0.00 (0.00–0.00)

Adjusted for age (continuous), sex (male/female), stage (I-II, III, IV), grade of differentiation (G1 or G2), somatostatin therapy (no/yes) and sample type (surgical resection specimen or biopsy).

## Data Availability

Anonymized data is available upon request from the corresponding author.

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
