# Peer review of "Monocarboxylate Transporters 1 and 4 and Prognosis in Small Bowel Neuroendocrine Tumors"

_cancers, 2022, doi:10.3390/cancers14102552_

Round 1

Reviewer 1 Report

Thank you for a well-written manuscript. 

Comments. 

1. Why is it interesting to measure MCT1 and MCT4?

2. What is the clinical significance of the study? Please add C-statistics and NRI/IDI among the statistical analyses. 

3. What does it add to current risk factors (KI-67 index, tumor stage, PS etc. etc.?)

4. NET G1 and G2 are very alike - maybe it would have been interesting to include NET G3 and NEC?

Author Response

Responses are attached as a separate Word file.

Reviewer 2 Report

In this study by Hiltunen et al, the authors have investigated how the expression levels of MCT1 and MCT4 correlate with prognosis and survival of SB-NT. Overall the data is interesting and the study is important. There are few comments that need to be addressed. 

The immunohistochemistry images could be made more visible and it would be helpful, if the authors could indicate the MCT1 and MCT4 positive cells.

The citation of figures in the text do no match with the actual figures. For instance Lines 194-199, the authors have mentioned Figures 3 and 4 but there are no actual figures. Please check this throughout the manuscript.

Author Response

(The authors gave the same response as above.)

Round 2

Reviewer 1 Report

Thank you for your revised manuscript which is improved. 

I am still not pleased with the statistical methods and recommend to at least include a ROC curve. 

Reviewer 2 Report

The authors have addressed all the comments
